# Collection of Silicon Detectors Mechanical Properties from Static and Dynamic Characterization Test Campaigns

Edoardo Mancini [1,2,*], Lorenzo Mussolin [1,3], Giulia Morettini [2], Massimiliano Palmieri [2], Maria Ionica [1], Gianluigi Silvestre [1], Franck Cadoux [4], Agnese Staffa [2], Giovanni Ambrosi [1], Filippo Cianetti [2], Claudio Braccesi [2], Lucio Farnesini [1], Mirco Caprai [1], Gianluca Scolieri [1], Roberto Petrucci [2] and Luigi Torre [2]

1. Perugia Department, National Institue of Nuclear Physics, Via A. Pascoli, 06123 Perugia, Italy; gianluigi.silvestre@pg.infn.it (G.S.); giovanni.ambrosi@pg.infn.it (G.A.)
2. Engineering Department, University of Perugia, Via G. Duranti 93, 06124 Perugia, Italy; filippo.cianetti@unipg.it (F.C.); roberto.petrucci@unipg.it (R.P.);
3. Physics Department, University of Perugia, Via A. Pascoli, 06123 Perugia, Italy
4. Nuclear Physics Department, University of Geneve, 24 Rue du Général-Dufour, 1205 Geneve, Switzerland
* Correspondence: edoardo.mancini@pg.infn.it

**Abstract:** Physics research is constantly pursuing more efficient silicon detectors, often trying to develop complex and optimized geometries, thus leading to non-trivial engineering challenges. Although critical for this optimization, there are few silicon tile mechanical data available in the literature. In an attempt to partially fill this gap, the present work details various mechanical-related aspects of spaceborne silicon detectors. Specifically, this study concerns three experimental campaigns with different objectives: a mechanical characterization of the material constituting the detector (in terms of density, elastic, and failure properties), an analysis of the adhesive effect on the loads, and a wirebond vibrational endurance campaign performed on three different unpotted samples. By collecting and discussing the experimental results, this work aims to fulfill its purpose of providing insight into the mechanical problems associated with this specific application and procuring input data of paramount importance. For the study to be complete, the perspective taken is broader than mere silicon analysis and embraces all related aspects; i.e., the detector–structure adhesive interface and the structural integrity of wirebonds. In summary, this paper presents experimental data on the material properties of silicon detectors, the impact of the adhesive on the gluing stiffness, and unpotted wirebond vibrational endurance. At the same time, the discussion of the results furnishes an all-encompassing view of the design-associated criticalities in experiments where silicon detectors are employed.

**Keywords:** silicon detectors; mechanical characterization; random vibrations; shock; mechanical space qualification





## 1. Introduction

At the present date, single-sided or double-sided silicon detectors (SSSD or DSSD) are widely utilized in space missions [1]; specifically, in experiments like AMS-02 [2–4] and DAMPE [5–7]. In addition, a substantial commitment has been made to the development of new and more efficient detectors, like mini-PAN [8,9], HERD [10–12], and ALADInO [13,14]. While, on the one hand, the research on the performance enhancement of the sensitive unit is of paramount importance, on the other hand, the overall experiment geometry is often as crucial for the development of improved detectors. Indeed, the performance-enhancement problem is faced at two scales: one deals with the active component of the detector, and the other with the experiment as a whole. Debating the latter, the critical parameters are hermeticity and non-active-material reduction. Within this framework, the scarcity of knowledge about detector mechanics could obstruct the maximization of such critical parameters. More extensively, the design of a silicon-based space experiment consists of

developing a stiff, strong, and thermally stable structure. The design process is an iterative balance between structural material minimization (opaque to the particles), mechanical integrity (during the launcher flight), and positional stability (during on-orbit thermal cycles). The design procedure is depicted in Figure 1, starting with an iterative phase where the design is proposed, structurally analyzed, and then updated, and followed by a production phase, a testing phase, and the final launch (in the ideal scenario where tests are successful and there is no need to iterate the design after them).

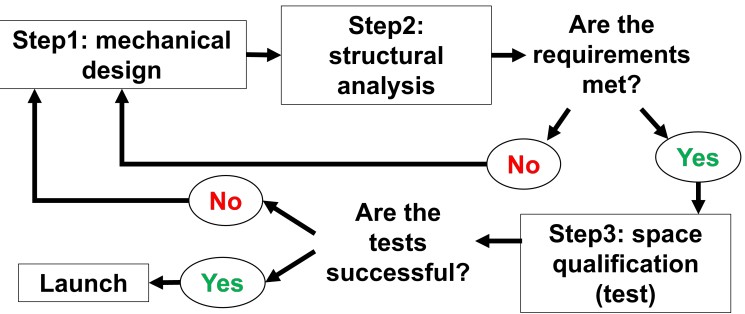

**Figure 1.** Standard design procedure.

The described procedure, which is standard in mechanical design studies (for more information on space structure design refer to dedicated literature [15,16]), has also been proved suitable for particle physics space experiments. Nevertheless, the unavailability of silicon-related mechanical data reduces model accuracy, and although missions are successful, they may operate with non-necessary structural mass. Indeed, in the design of the supporting structures, the contribution of the silicon to the system's mechanical properties is often neglected and only the contribution in mass is considered. Instead, considering the silicon mechanical properties by accounting for their stiffness and structural properties, or exploring a precise knowledge of silicon failure parameters, could open new and fascinating possibilities (like the one presented in [17]). Adding this concept to the design procedure leads to the new approach presented in Figure 2.

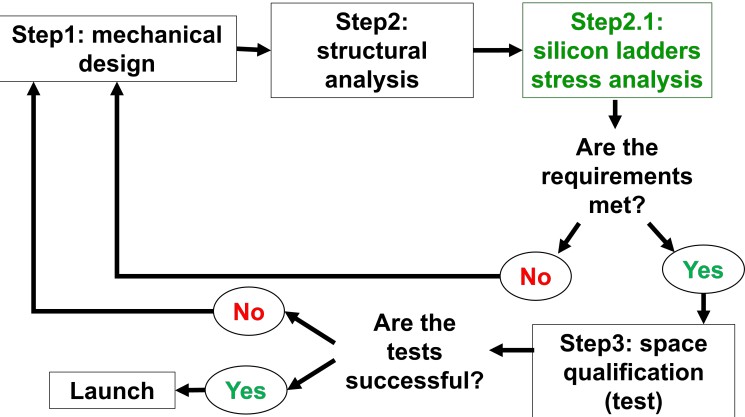

**Figure 2.** Procedure involving silicon detector properties.

To further clarify, structural analysis targets the provision of a priori information about an experiment's mechanical behavior during the flight and its operative life. As for any other model, a higher level of detail means more accurate results. With structural integrity being mandatory for the mission's success, the designers fill the gap between model and reality with safety margins, hence additional mass.

Conversely, introducing a proper silicon description into the model could reduce the uncertainties (ergo the margin), resulting in a mass reduction and performance increase. To build better models, two kinds of data are needed: stiffness-related and strength-related.

The first are important to account for the presence of detector assemblies, not only as additional masses but as mechanical objects. The second are to predict the detector's failure conditions more precisely. Once again, the need is for information on the whole payload and not only the sensors; the analysis should embrace the whole system installed on the structure. Consequently, to provide a comprehensive analysis, the present document does not only dwell on the characterization of silicon tile stiffness and strength (presented in Section 2) but also discusses the role of the structure–detector mechanical interface (glue) in the dynamics (in Section 3), and finally also on the wirebonds, which are crucial for the detector's functionality (Sections 4.1 and 4.2).

In conclusion, the present work is a comprehensive study of silicon detector assemblies' mechanical aspects. The work breaks this into three main parts. The first concerns a set of flexural tests performed on silicon detector tiles to gain knowledge of the stiffness and strength properties of the detective material. The second presents studies testing the impact of different glues (specifically silicon-based and epoxy-based) on the dynamic response of xSSDs. The third evaluates the vibrational tolerance of wirebonds. The wirebonds analysis is broken into two parts: a report of pull test activities, ensuring the quality of the micro bonding manufacturing process; and the proper set of high-level random vibrations introduced before.

An in-depth analysis of all the presented aspects can not be summarized in a single work, and this is not the intention. Instead, this research intends to provide a holistic view of the problem and collect the authors' experiences on the matter. This aim is attained through the provision of useful data and the discussion of possible design issues. As a whole, this paper introduces various topics paving the way for future investigations.

*Silicon Detector Description*

This section briefly describes silicon detector assemblies, with details relevant to the presented studies.

Silicon detectors are widely used in particle physics and represent a valid means of obtaining information in this area. Their main constituent is doped silicon enriched with a superficial metalization on both sides. In the case of SSSDs, one side shows a continuous metalization, while the other has conductive stripes in correspondence with implanted doped silicon areas. The case of the DSSD is more complex (please refer to the dedicated literature [18,19] for a comprehensive description). An SSSD is shown in Figure 3.

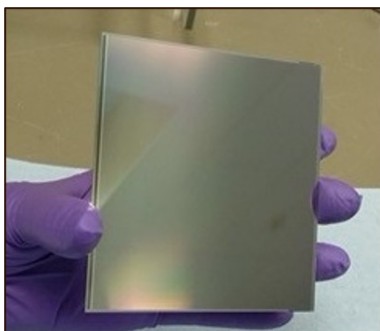

**Figure 3.** Single-sided silicon detector or tile .

The silicon dimensions are in the order of 100 mm. Therefore, to cover large areas, it is necessary to use more than one SSSD, or tile. Commonly, an active plane is created by placing sub-assemblies, called *ladders*, one next to the other. A ladder is a line of sensitive tiles electrically connected one to another. Figure 4 depicts a ladder. The silicons in the figure are 97 mm $\times$ 97 mm. A PCB is visible at the end of the silicon ladder, it carries the readout ASICs and interfaces to the off-detector electronics.

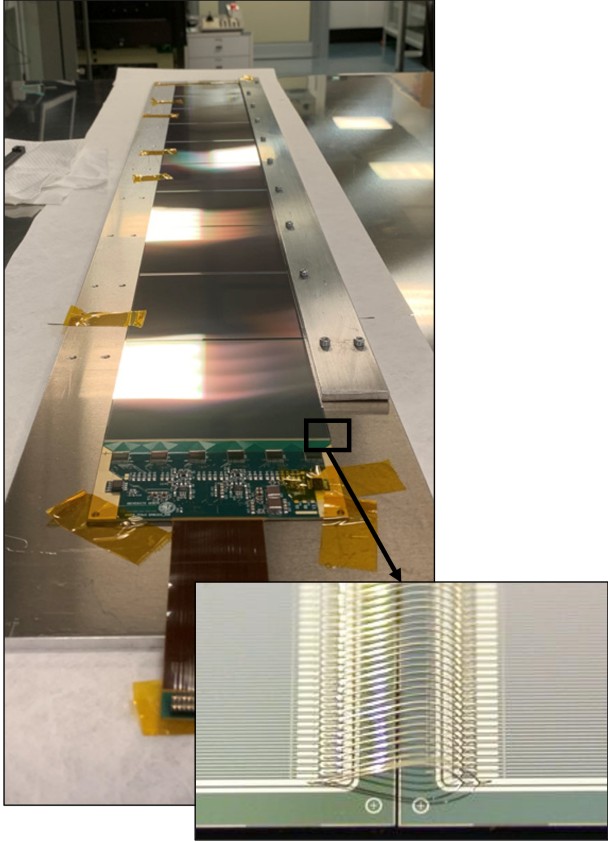

**Figure 4.** Ladder.

Figure 4 introduces the ladder and the wirebonds (on the left); i.e., the electric connections between adjacent tiles and between tiles and electronics. These tiny soldering junctions are how tiles are electrically connected to the read-out electronics (or *front end*). The "micro" nomenclature derives from the dimension of the wire used in the connection. The said dimension is contained by the distance between adjacent strips (each of which has to be connected to a single electronic channel), ranging from tens to hundreds of μm. The touching of wires results in measurement failure. Therefore, the size of the wire used for the connections should be comparable to the strip pitch. The diameter of the aluminum wire used is 25 μm, and a specific machine completes the attachment. Wirebonds are extremely fragile and break if touched or pulled. Thus, the silicon tiles and the front-end electronics are glued together on a substrate. The substrate exploits electrical duties by conducting the bias voltage from the electronics to the bottom of each detector. Hence, it is necessary to provide both a mechanical and electrical interface between the tiles/electronics and the substrate. In other words, the glue layer connecting the silicon and the PCB to the substrate should be at least partially conductive. Thus, a conductive and a structural glue must be employed to ensure electrical connection and adhesion.

Finally, by placing multiple ladders, it is possible to create large active surfaces.

## 2. Silicon Detector Mechanical Characterization

The scope of the present section includes the mechanical characterization of the silicon detector material. The information of interest relates to stiffness and strength. Therefore, the retrieved parameters are the Young's modulus and the maximum stress and strain bearable by the silicon. In the present discussion, the material is thought to be isotropic and homogeneous (for a more detailed discussion on silicon directional properties please refer to [20–22]).

### 2.1. Test Samples

The test batch consisted of fourteen DSSDs with dimensions of $72.00 \times 41.40 \times 0.30$ mm$^3$ (measured with 0.01 mm of accuracy). The crystal orientation was <111> for all specimens. The tested samples were spares from the AMS-02 experiment and therefore fully representative of real space hardware.

### 2.2. Test Description and Execution

Each sample underwent a *three-point bending* test, consisting of the application of a force perpendicular to the silicon surface while the sample rested on two supports. The force and the supports were round bars capable of exerting forces but not moments, thus constraining the perpendicular displacement but not the rotation. The test is detailed in Figure 5. The machine used for the test was a LLOYD LR30K.

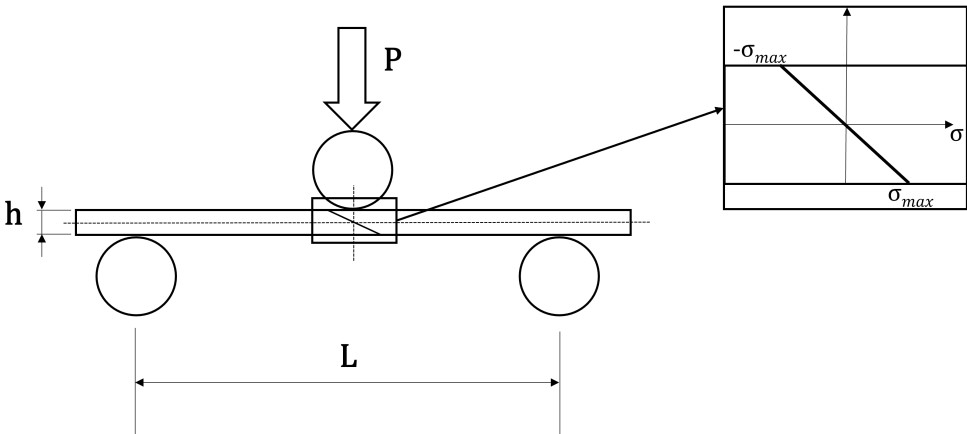

**Figure 5.** Three–point bending test scheme.

The illustration also presents the through-the-thickness stress profile, linked to the strain profile through the silicon elastic properties. The stress profile was induced in the SUT by the applied load $P$. Equation (1) relates the input force to the surface stress.

$$\sigma_{surf} = \frac{3PL}{2bh^2} \tag{1}$$

Concerning Figure 5, $P$ and $\sigma_{surf}$ are the applied force and the resulting surface stress, respectively, $h$ is the specimen thickness, $b$ is the specimen's width (through-the-paper dimension), and $L$ is the free span (distance between supports), equal to 50 mm in the present case. Figure 6 portrays the test setup.

Starting from the rest position ($P = 0$ N), the equipment moved the central rod downwards and measured the force opposed by the system under test (SUT). The motion had an initial engagement phase, where the rod filled the gap between the resting position and the SUT surface. This was followed by a strain test, where the force applied by the moving gauge increased, compensating for the opposing force of the deforming specimen. Eventually, the sample broke, and the force applied by the gauge dropped to zero. Using Equation (1), the stress in the silicon sample was extracted from the measurement of the gauge and was plotted in the time–stress graph shown in Figure 7. The circumstance that the stress increased only linearly indicated that the silicon only deformed elastically until the breaking point. The stress at the breaking point for this sample was 208.5 MPa.

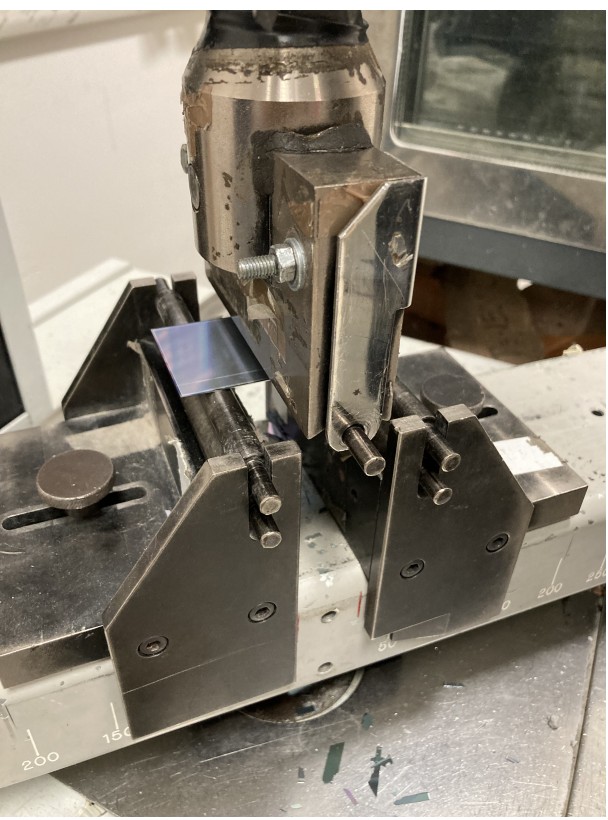

**Figure 6.** Three-point bending test setup.

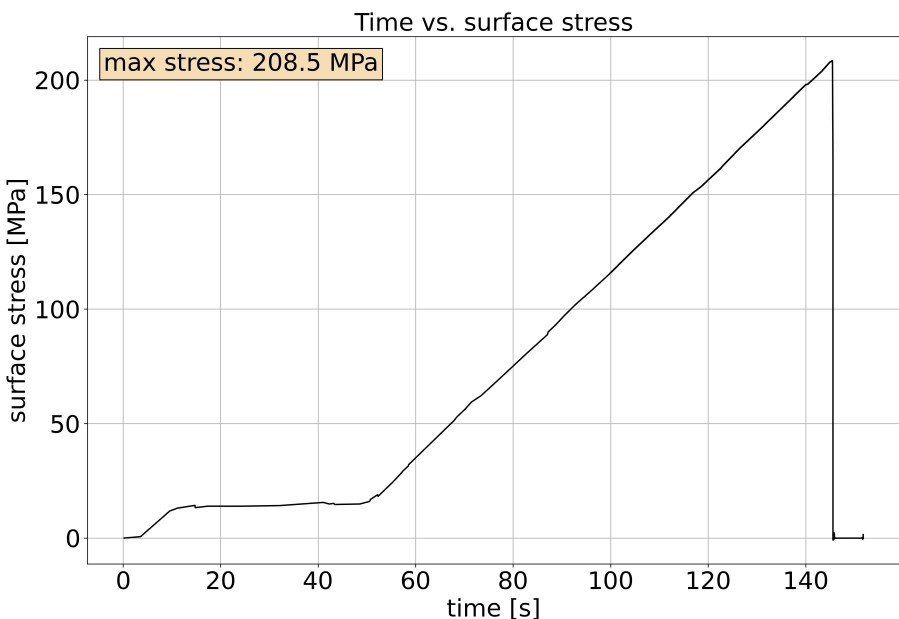

**Figure 7.** Silicon detector surface stress as a function of time.

The reported maximum stress values are presented in Table 1, while the strains are in Table 2.

**Table 1.** Sample data for max stress.

| Sample | Max Stress [MPa] |
|---|---|
| 1 | 187.4 |
| 2 | 305.1 |
| 3 | 272.8 |
| 4 | 220.2 |
| 5 | 384.3 |
| 6 | 236.3 |
| 7 | 208.8 |
| 8 | 293.6 |
| 9 | 151.5 |
| **Max** | **384.37** |
| **Min** | **151.6** |

**Table 2.** Sample data for Max Strain $\left[\frac{m}{m}\right]$.

| Sample | Max Strain $\left[\frac{m}{m}\right]$ |
|---|---|
| 1 | −0.0015 |
| 2 | −0.0019 |
| 3 | −0.0010 |
| **Max** | **−0.00102** |
| **Min** | **−0.0019** |

Having calculated the maximum stress bearable by the single tile, the focus shifted to the stiffness relating stress and strain through the relation of Equation (2).

$$E = \frac{\Delta\sigma}{\Delta\epsilon} \tag{2}$$

Since $\sigma$ can be computed from the applied force, it was necessary to measure $\epsilon$ to determine $E$. For this reason, mono-directional strain gauges were installed on eight out of the fourteen specimens. Of the former, three samples were tested to failure, with five in the elastic region, thus allowing multiple repetitions on the same specimen (to verify the repeatability of the measurement). Finally, the Young's modulus was extracted from the strain–stress curves, like the one in Figure 8. The measured data are reported in Table 3.

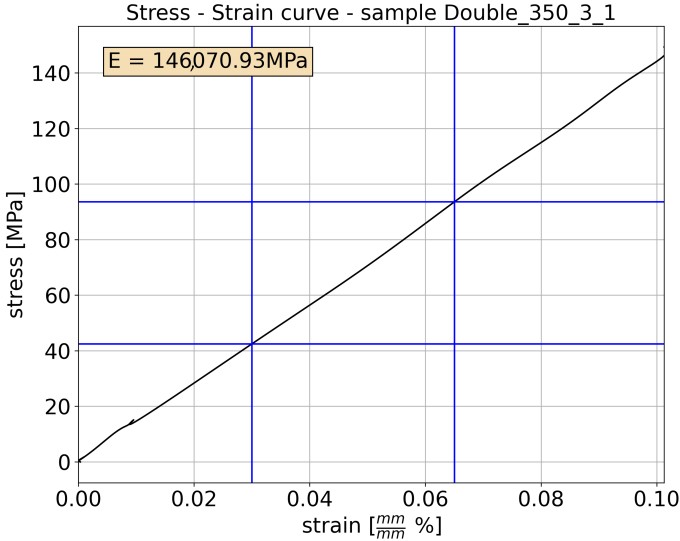

**Figure 8.** Stress–strain curve and Young's modulus estimation.

**Table 3.** Sample Data for E [MPa].

| Sample | E [MPa] |
|--------|---------|
| 1 | 142,996.0 |
| 2 | 151,949.7 |
| 3 | 131,619.0 |
| 4 rep1 | 159,680.2 |
| 4 rep2 | 151,574.8 |
| 4 rep3 | 154,870.3 |
| 4 rep4 | 159,155.2 |
| 4 rep5 | 151,675.6 |
| 5 rep1 | 135,986.4 |
| 5 rep2 | 139,668.3 |
| 5 rep3 | 135,712.0 |
| 5 rep4 | 134,508.5 |
| 5 rep5 | 139,795.8 |
| 6 rep1 | 121,552.2 |
| 6 rep2 | 124,688.9 |
| 6 rep3 | 121,231.5 |
| 6 rep4 | 116,789.3 |
| 6 rep5 | 118,367.0 |
| 7 rep1 | 146,106.7 |
| 7 rep2 | 146,901.6 |
| 7 rep3 | 145,125.7 |
| 7 rep4 | 151,382.1 |
| 7 rep5 | 150,104.5 |
| **Max** | **159,680.2** |
| **Min** | **116,789.3** |

Finally, another batch of 11 DSSD specimens, identical to the ones used for these tests, were weigh on a scale with accuracy 0.1 g. Through which it was possible to estimate the DSSD density (through the notorious Equation (3)).

$$\rho = \frac{m}{V} \tag{3}$$

### 2.3. Test Output Summary

In conclusion, the test campaign used fourteen silicon tile specimens, eight of which were equipped with strain sensors. The testing machine provided force data that were easily related to the through-the-thickness stress thanks to Equation (1). On the other hand, the strain measurements and Equation (2) led to the estimation of the Young's modulus and the maximum strain (for the three cases in which failure occurred). Additionally, the average density was computed through the weight measurements on eleven sensors. The test results are summarized in Table 4.

**Table 4.** Summary of silicon experimental campaign results.

| Property | Mean ($\mu$) | Standard Deviation ($\sigma$) | Sample Size |
|----------|--------------|-------------------------------|-------------|
| Density $\left[\frac{\text{kg}}{\text{m}^3}\right]$ | 2392 | 69.5 | 11 |
| Max Stress (MPa) | 251.15 | 70.60 | 9 |
| Max Strain (%) | 0.1457 | 0.0430 | 3 |
| Young Modulus (GPa) | 142.19 | 10.19 | 8 (23 reps.) |

### 3. Dynamic Effect of the Adhesive Bond to the Structural Substrate

*3.1. Overview of the Design Challenge*

The present section discusses the effect adhesives have on the stresses experienced by silicon sensors when subjected to vibration. From the author's experience, two different classes of adhesives can be used for structural purposes: epoxy-based or silicon-based. The first being very rigid and the second more compliant. As explained later in this section, the second are more suitable for this application, since they introduces damping at the interface and mitigate shock loads. Nevertheless, the adhesive effect is different for each detector configuration, and some may be more susceptible to certain specific effects than others. A further complication is the requirement of a second, conductive glue, providing the bias to the backside of the sensors. This has to be epoxy-based and must be applied with a minimum area, to meet the electrical conductivity requirement. A stiff adhesive runs against the desire for the glue to dampen the transfer of vibrations. The challenge is to find a design combining a conductive glue with a softer silicon-based glue.The variable design parameters include the relative areas, the applied patterns, and the thickness of the glues, each of which is limited by further constraints

The criticality of the adhesive choice emerged during the shock test of the DAMPE [23] quarter plane prototype shown in Figure 9, where the tiles broke.

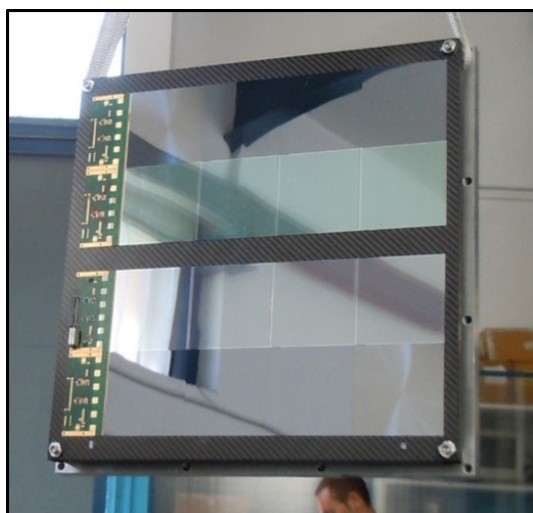

**Figure 9.** DAMPE quarter plane prototype.

In the present situation, the ladders, shown in Figure 9, were connected to the plane with the glue pattern shown in Figure 10.

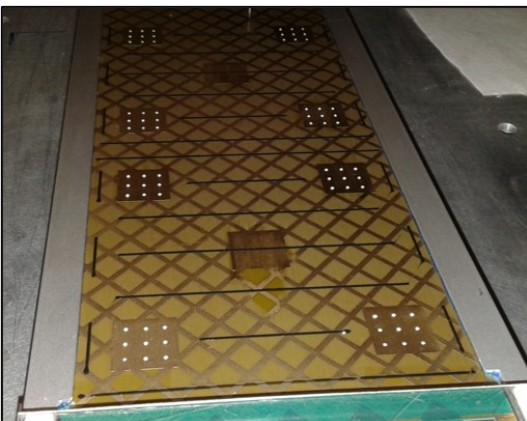

**Figure 10.** DAMPE gluing pattern.

The reader can notice two different adhesive patterns: thin dark lines of epoxy structural glue (not to be mistaken with the bottom brown copper cross pattern) and gray dots of electrically conducting glue (necessary to ensure the system's functionality).

As previously stated, epoxy glues are stiff and ensure a rigid connection between the ladder substrate and the supporting plane but at the same time provide little damping. The stiffness of the connection was considered the prime factor responsible for the failure registered during the shock test of the prototype (depicted in Figure 11).

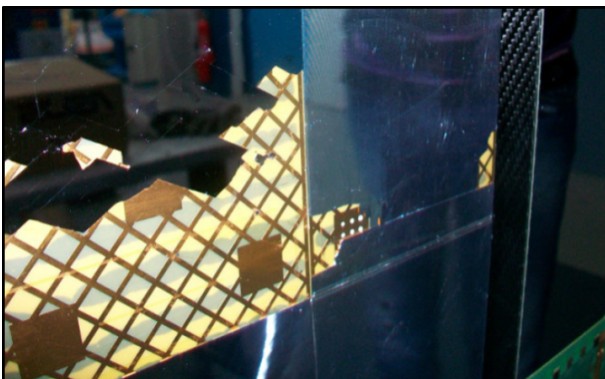

**Figure 11.** Silicon detector failure during the space qualification shock test.

For this reason, in the DAMPE flight model, structural duties were fulfilled by a compliant glue (silicon-based), replacing the much stiffer epoxy-based adhesive. For the sake of a better understanding, the difference in Young's modulus ($E$) of the two adhesives is very relevant, going from the 1 GPa of the epoxy-based to the 1 MPa for the silicon-based.

The successful launch and correct on-orbit operation of the DAMPE proved that this choice was effective. After this result, no further studies were performed and the silicon-based glue was made the standard for the adhesion of tiles to substrates.

Recently, it has been observed that the best electric glue used for non-structural purposes is an epoxy-based adhesive.This observation motivated a new study on the effect of the latter on tile mechanics, and this is the subject of the present section. To clarify, the change of structural glue was sufficient for the mentioned design. Although the conductive glue was still present, the performance increase resulting from the change was sufficient to pass the qualification test. Indeed, the amount of conductive glue was less than that of the structural glue. As a whole, the specific design was successful and no further investigations were performed. Conversely, the present study reopens this topic, to highlight this issue and provide general information to help future designs.

### 3.2. Glue Data

Before moving on with the test campaign, it is interesting to provide information on the gluing.

Starting with the involved adhesives, the epoxy glue used for this work was 3M scotch-weld epoxy adhesive 2216 gray, the Si-based was Dow Corning 3145 RTV MIL-A-46146 Gray, and the conductive glue was EPO-TEK® EJ2189. The glue was using a syringe. The deposition patterns were straight lines for structural glue (either Si or epoxy-based) and single dots for the conductive glue (as per Figure 10). The glue thickness was about 30 μm.

### 3.3. Experimental Campaign

The continuous effort towards the development of more efficient and/or compact detectors has led to new geometries, where the issue presented in the above can become relevant. Specifically, a geometry such as that of a mini-PAN with a single active surface, adhering to the substrate only on its boundaries, could increase epoxy-associated effects. Concurrently, this scenario is perfectly suited for tackling the topic.

Leveraging the former discussion, a test campaign was set up. The test samples were two mini-PAN trackers installed on the same mechanical interface (tracker module version 1) and then vibrated. The first was a tracker PCB with a dummy silicon detector (mechanically equivalent) glued only using structural glue (from now on, we will refer to the silicon-based glue as *structural glue*). The second was like the first (even the same PCB is used) with a different gluing: now both the structural and conductive adhesives were used. Both tests employed mechanical dummies. The detectors were equivalent to real ones mechanically but not electrically. Hence, there was no need to apply the bias(the conductive glue added for the bond to be mechanically relevant, not for electric purposes). Except for the adhesive bond, the two tests were the same. Hence, the following discussion applies to both.

The goal of the test was the estimation of the dynamic effect of the glue. In practice, by monitoring the overall damping of the assembly, it was possible to appreciate the effect of the adhesive.The damping was a good control parameter for our goal. Indeed, this study aimed to show the criticality of the glued connection and to qualitatively show the extent of the changes. A comprehensive discussion would require an extended discussion, foreseen for future studies. Instead, here, the authors would like to provide one lesson learned to guide designers facing similar issues.

It is important to attest that high-damping has a very beneficial effect on non-static loads and especially on shock loads [24]. There are several ways to compute the damping, most of which are experimental (that is the reason why this study relied on experiments) [25,26]. In this case, the choice was the method the commonly called the *3 dB method*. This method predicted the damping estimation from the frequency response function (FRF) peak amplitude drops [27,28]. Here, the damping of the first peak was considered (being associated with the flexural mode of the silicon tile). Thus, the damping value was a direct measure of how the glued interface filtered the inputs applied to the PCB (the adhesive was responsible for the load transfer between the PCB and the silicon). More information on the FRF is provided in the literature [29–31]. In the ideal case of undamped structures, the FRF amplitude at the peaks would be infinite (asymptotes in the function). In reality, all mechanical systems are damped and the FRF amplitude is finite. It can be proved that the damping is proportional to the peak width; narrower peaks have higher damping than wider ones. The *3 dB method* was used to estimate the damping from the peak aperture. For this aim, Equation (4) was used, where $\Omega_0$ is the peak frequency, and $\omega_1$ and $\omega_2$ are the frequency points to the left and right of the peak with an amplitude 3 dB less than that of the peak (Figure 12).

$$\xi = \frac{(\omega_2 - \omega_1)}{2\Omega_0} \tag{4}$$

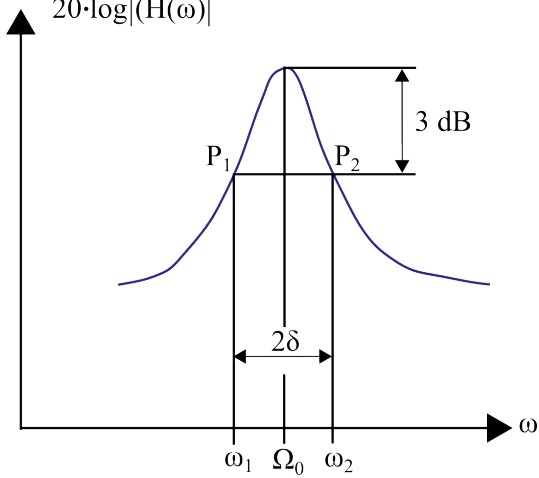

**Figure 12.** Illustrated explanation of the *3 dB method*.

Leveraging this method, it was possible to estimate the damping for both configurations. The analyzed cases were explanatory conditions chosen to present the issue and provide some indicative numbers. This part of the research did not provide a comprehensive report on silicon gluing. Indeed, it would be necessary to perform specific research considering different patterns, glues, and geometries. This is, of course, a prospect of the study. Conversely, the present research aimed to depict a serious mechanical issue to be considered during the design phases and to build a basis for more in-depth studies.

Detailing the test execution, both SUTs were installed on a shaking table and underwent a frequency sweep. To avoid any ambiguity, a shaker table is an experimental apparatus capable of generating dynamic loads on a platform called the *shaker head*. The provided load can be random or harmonic [32]. The first consists of a time-varying signal constituted by the superimposition of multiple sinusoidal functions with different frequencies and random phases. The second consists of the application of harmonic signals with fixed amplitude and time-varying frequencies, according to a predefined time-frequency law named the sweep-rate [33]. The main function of the shaker is to apply reference loads to verify the capability of the SUT to withstand a given load profile (either random or harmonic). Nevertheless, the shaker can also be used to determine the FRF at a certain point of the SUT. Once more, the FRF is the relation between the input (provided by the shaker in this case) and the output signal in a control point. If the structure is perfectly rigid, the measured signal is identical to the input signal and the FRF is 1 at all frequencies. Conversely, elastic structures resonate, thus the measurement at the resonance frequency will be higher than one (and dependent on the damping). As a whole, in this part of the research, the shaker was used to determine the FRF between the shaker profile and the center of the silicon. To do so, a low amplitude (to avoid damage to the structure) harmonic load was applied. The shaker used for the test was a Sentek L0315 reference.

Figure 13 gives a snapshot of the test setup.

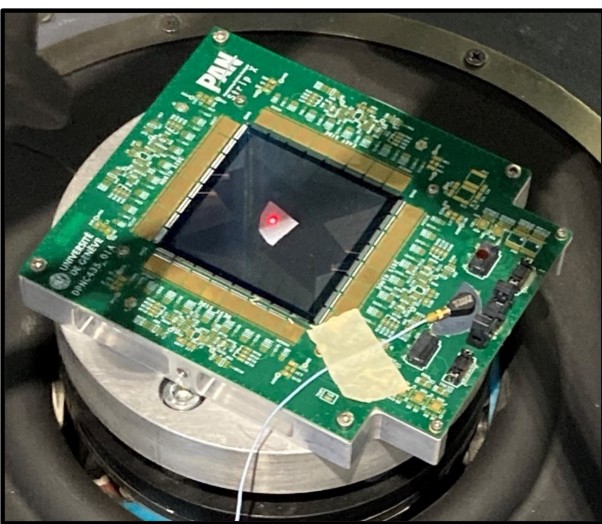

**Figure 13.** Damping estimation setup.

Here, the shaker head (metallic disk below the mini-PAN tracker fixture) and two measurement points are visible: one in correspondence with the accelerometer, and the other a red dot. Indeed, while the first measurement was taken using standard techniques (piezoelectric accelerometer), for the second measure, a laser interferometer was employed. This choice was motivated by mass considerations: sensors should not affect the dynamics of the SUT. As a rule of thumb, the accelerometer should be 100 or 1000 times lighter than the tested component. Given the mass of the silicon surface, even the lightest accelerometer (0.2 g) would affect the results. For this reason, a laser interferometer was used. It is worth highlighting once more that there was a hole in the PCB in correspondence with the silicon surface, and only the sides of it were in contact with the PCB.

The final result of the study is presented in Figures 14 and 15, respectively, illustrating case 1 (only structural glue used) and case 2 (nominal bond: structural and electrical glue used).

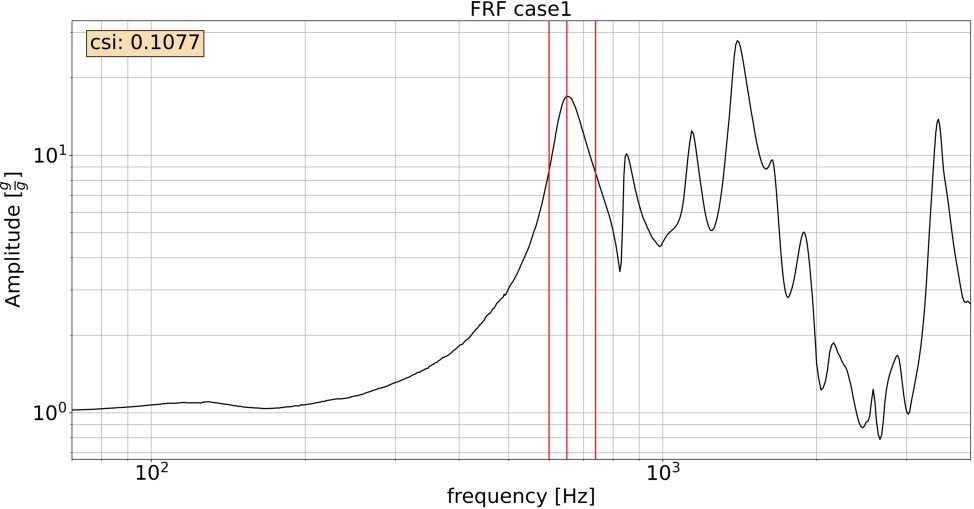

**Figure 14.** FRF and damping estimation case 1.

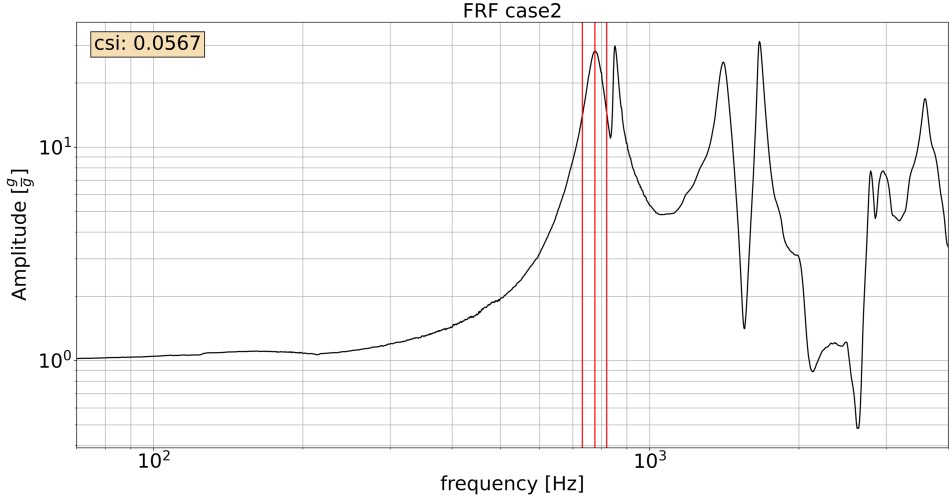

**Figure 15.** FRF and damping estimation case 2.

In both cases, the input profile was harmonic, with an amplitude of 4.905 $\frac{m}{s^2}$ ($\frac{1}{2}$ g), spanning between 20 Hz and 3.5 kHz. As specified before, the analysis focused on the first peak, which is considered the most relevant for the present discussion.

Discussing the results, the difference in damping was far from negligible. Indeed, the damping dropped from the $\xi = 0.11$ of the silicon case to the $\xi = 0.056$ of the stiffer condition. Thus, the test reported a damping drop of about 47%, proving that the glue effect on this kind of mechanical bond is far from negligible.

To conclude, the present study intended to stress the criticality of xSSD gluing, first noticed during DAMPE prototype shock tests. To attain this goal, two identical mini-PAN trackers were tested. The SUTs were identical in every aspect, except the adhesive configuration: one employed only silicon-based glue, and the other used both silicon-based and epoxy-based (necessary for electrical purposes) adhesives. The dynamic effect of the different gluing configurations was assessed through the measurement of the first-peak damping from the experimental FRFs. Given the contained mass of the silicon, it was necessary to perform the FRF measurements with a laser interferometer.

The result leads to the conclusion of the importance of gluing for the detectors' dynamics and the necessity of reducing the amount of conductive glue as much as possible if the objective is to have a soft and damped bond.

## 4. Wirebond Mechanical Studies

This part of the research concerns the mechanics of wirebonds. The discussion is split into two: an initial part collecting data from a wire pull test campaign, and a vibration campaign. The goals of the first were the assessment of wirebond connection quality and the sharing of experimental data with the scientific community. The goal of the second was to extend the heritage (coming from various successful space missions) of wirebond vibration endurance to more general cases. Indeed, the literature [34] advises the encapsulation of wirebonds, while various space missions have successfully employed unencapsulated connections. To conclusively prove the space suitability of unencapsulated bonds, the present study performed a random vibration campaign. Three different samples were vibrated at levels far above the vibrational space qualification levels requested by space standards. Additionally, a preliminary shock campaign was performed. The shock test aimed to enrich the picture and provide a more solid result. In any case, the latter can not be considered a nominal shock qualification campaign, because the required levels were not attained. Revisiting the earlier discussion, wirebonds are the electric connections between silicon tiles or between silicon tiles and front-end electronics. For the experiment to function properly, connections could not brake nor short-circuit (touching the surrounding wires). Thus, the dimensions of the connection were constrained by the spacing and the width of the strips. This necessitated the employment of 25 µm diameter wires (99% purity). The bonding procedure required a specific apparatus. Figure 16 presents a picture taken during the bonding process (bonding tool on the right, pale yellow).

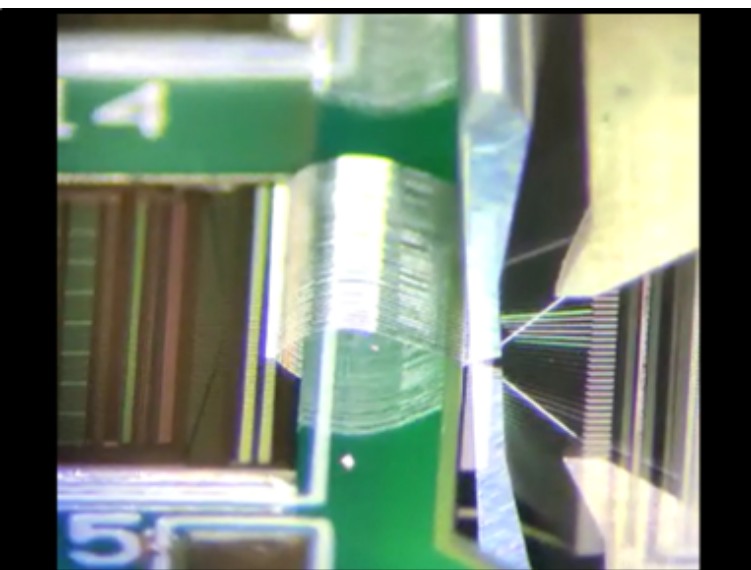

**Figure 16.** Bonding process.

### 4.1. Manufacturing Process Verification

To verify the mechanical strength of the electric connections, a set of 515 samples was tested. Each wire was pulled using a custom-made hooked dynamometer. The acceptance criteria mandated that only 15% of failures were padlifting. In the remaining cases, the wire broke either in the middle or at the base.

Detailing the setup, Figure 17 shows a picture taken during the test, whose structural configuration can be summarized with the scheme of Figure 18.

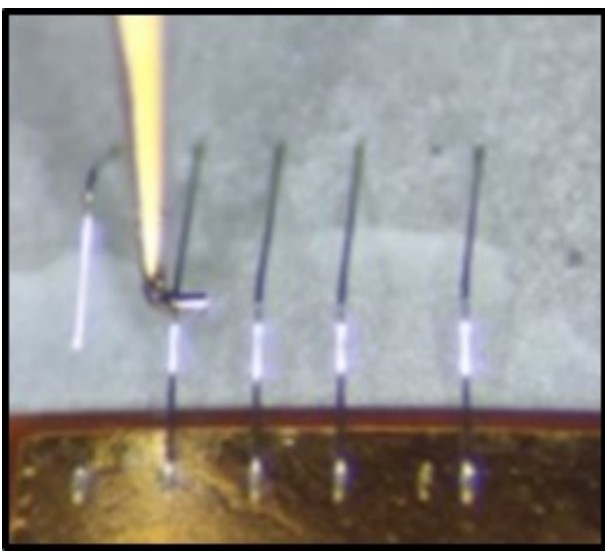

**Figure 17.** Wirebond pull test—test picture.

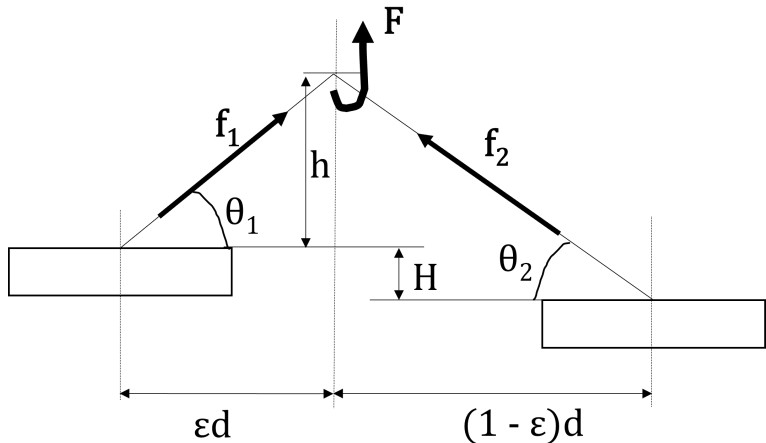

**Figure 18.** Wirebond pull test—structural scheme.

The latter presents the general situation of this kind of test. For the specific case, the parameters were

- $h = 750$ μm
- $H = 0$ μm
- $d = 1500$ μm
- $\epsilon = 0.5$

Resulting in $f_1 = f_2 = F\frac{\sqrt{2}}{2}$. The assumption for $\epsilon$ is quite strong: the application point depends on the operator's skills. Nevertheless, given the numerosity of the sample, we can assume that the uncertainty on $\epsilon$ was statistically mitigated.

The retrieved data are presented in Table 5.

**Table 5.** Summary of wirebond pull test data.

| Property | Mean ($\mu$) | Standard Deviation ($\sigma$) | Sample Size |
|---|---|---|---|
| Pull Force at break point $[gf]$ | 12.47 | 1.80 | 515 |

### 4.2. Vibrational Tolerance of Wirebonds

The majority of mechanical load experienced by a space mission is launcher-associated. Indeed, the on-orbit placement phase is by far the most critical for space systems.

Once again, the loads experienced in this phase are mainly dynamic and can be split into three categories: harmonic loads, random loads, and shock loads. The first comes from the launcher resonances; the second from the non-deterministic inputs from acoustic, aerodynamic, and thrust generation apparatus; and the third from instantaneous phenomena, such as booster and launcher stage separation.

For an object to be space qualified, it has to be subjected to a mechanical qualification campaign. Such campaigns involve the application of test loads with levels dependent on the specific case; based on the experiment mass, the configuration and position in the loads experienced during the flight can be very different.For random tests, common practice involves the use of a standard (American [35] or European [36]) or a launcher user manual (e.g., Falcon's [37]) for the profile definition. Instead, here, the test profile was intentionally more severe than that of the standards, to account for the possible dynamic amplification of the experiment structure. The study aimed to demonstrate the extreme tolerance of wirebonds to vibration and to prove the suitability of unencapsulated wirebonds for space applications. The second objective was quite relevant for the authors. Although the literature [34] suggests encapsulating the bonds for mechanical protection, this specific application experience proved this to not be mandatory; the space operative missions discussed in the introduction did not employ this solution. In any case, bond geometry severely affects the mechanical behavior, and success may be related to the specific cases. Hence, a dedicated analysis was deemed necessary to avoid future problems and to qualify the naked-wire approach.

To attain this goal, the SUTs were intentionally overtested. Not knowing the launch configuration, the objects were tested in different directions (X, Y, and Z). High-level random tests were the primary objective of this activity, and z-directed shock tests enriched the overall picture. In conclusion, the present activity aimed to cover all possible flight conditions, provide data applicable to future experiments, and definitively prove the space suitability of unpotted wirebonds.

### 4.2.1. SUTs and Setup

To produce general results not associated with specific configurations, three different test samples were selected. The first sample (Figure 19) comprised two AMS-02 DSSDs spaced 11 mm apart. It was not possible to electrically verify the object; thus, visual inspection was the only criterion available to judge the test results. The bond was intentionally longer for this campaign, to validate a sort of worst-case scenario. The detectors were glued on an FR-4 substrate and then on an aluminum plate (interface to the test apparatus).

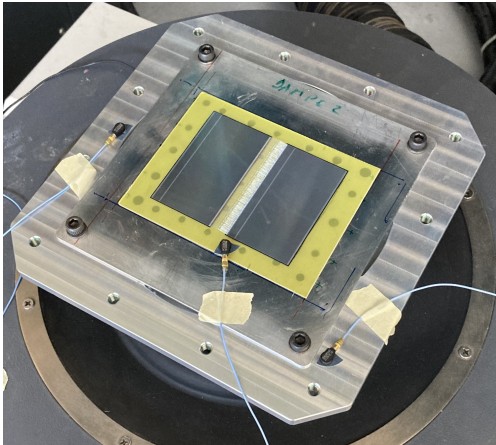

**Figure 19.** Test sample 1 *long bonds*.

Test sample no. 2 was a tile and electronics functional assembly from the AMS-02 L0 upgrade. Figure 20 presents the SUT 2. The test started with two pairs of front-end chips (the external ones from each side) not connected to the silicon sensor surfaces. Similarly to test sample 1, the assembly was glued on a metallic plate, working as a test interface.

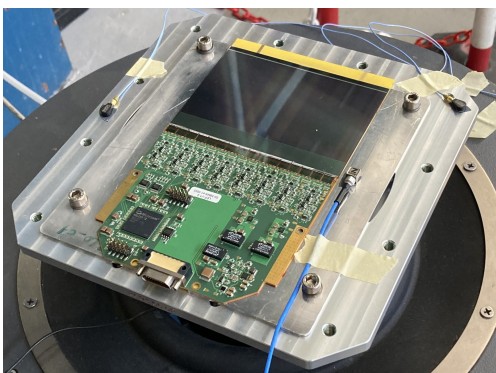

**Figure 20.** Test sample 2 AMS-02 L0 upgrade.

Test sample no 3 in Figure 21 employed a DAMPE silicon and front-end and it functioned like SUT 2. For all SUTs, silicon-based glue ensured the adhesion of the SUTs to the mechanical interface and the bonds are naked (no potting).

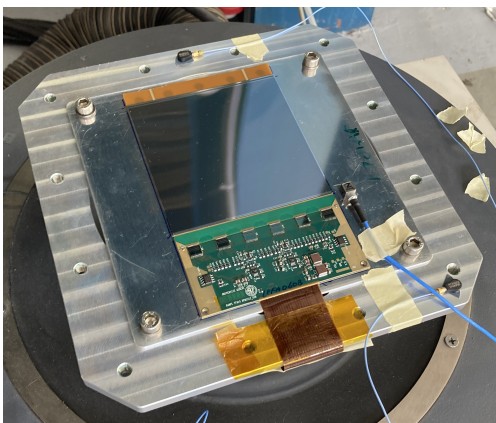

**Figure 21.** Test sample 3 DAMPE detector.

Each assembly underwent a random vibration along the X, Y, and Z axes (where Z is the direction normal to the silicon surface). No harmonic testing was performed, because the sine sweep space qualification excites frequencies below 100 Hz and in this range the tested objects behaved like rigid bodies; there was no dynamic amplification and the load experienced equaled the input. Instead, on higher portions of the spectrum, the component's dynamics played a relevant role, and the input loads were amplified. This is the critical part of the spectrum.

The random test profile ranged from 20 Hz to 2000 Hz with a plateau between 100 Hz and 600 Hz and a ramping profile elsewhere (the same shape as the one presented in Figure 22). The profile is provided in terms of PSD, as in the cited standards [35–37]. The interested reader can find additional information on the topic in the literature [15,29].

The severity of the random profile was measured through the root mean square (RMS) acceleration. All samples were tested at different levels (up to the maximum permitted by the apparatus i.e., 40 g RMS along the Z axis and 17 g RMS along X and Y) in the three directions. Concerning the equipment, the testing apparatus was the same Sentek L0315 shaker as in Section 3.3, with a different head expander for the X, Y, and Z tests. Figure 23 presents the X–Y test head expander, while Figure 24 presents that for Z.

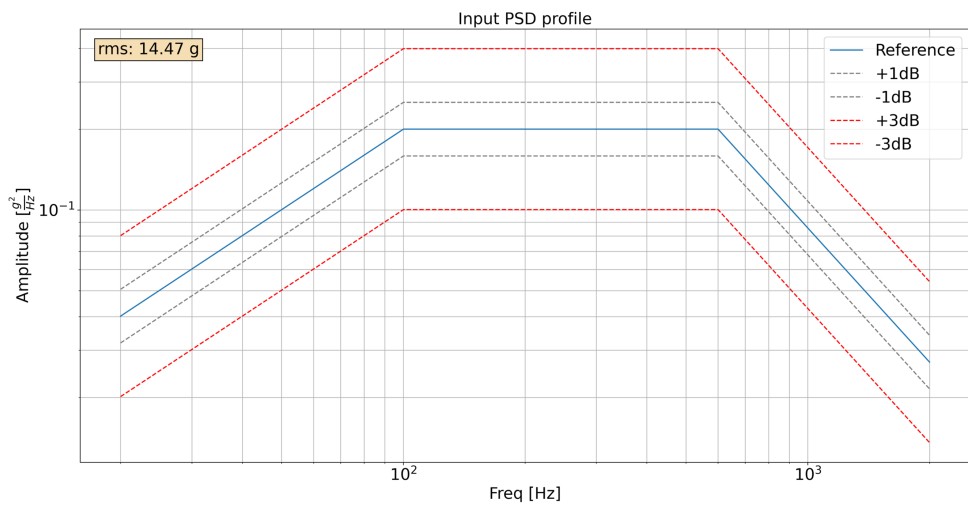

**Figure 22.** Random test profile of power spectra density from reference [35].

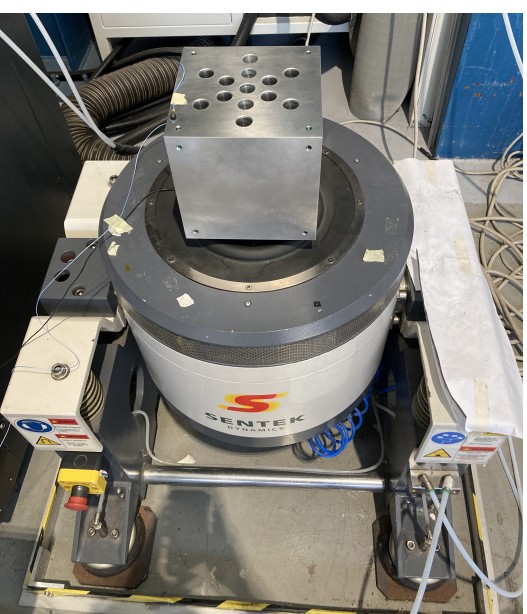

**Figure 23.** Head expander used for the X and Y tests.

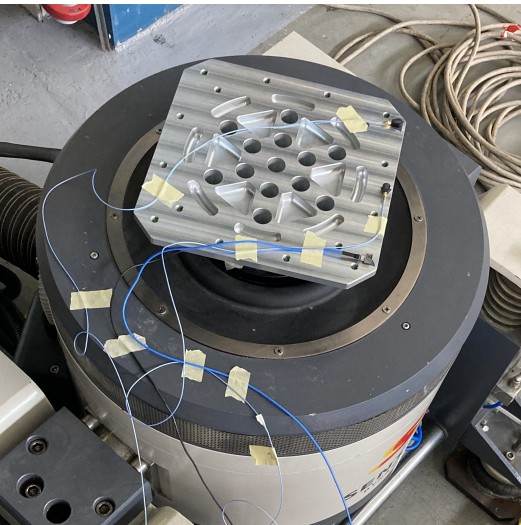

**Figure 24.** Head expander used for the Z tests.

The Z fixture was necessary due to the dimensions of the assemblies, especially for test samples 2. Instead, the X–Y cubic fixture allowed the testing in the specified direction with a shaker force directed along Z (the SUTs were attached to the lateral face of the cube). Additionally, on-shaker Z-direction shock tests were performed on all samples.

### 4.2.2. Wirebond Vibration Tests

Extending the previous discussion, the campaign involved two types of test: random and shock (harmonic was not considered relevant since it excites lower frequencies). Random tests were performed in X, Y, and Z configurations. Shock tests were performed only along the Z direction. Concerning the test configuration, the Z setups for all test samples are visible in Figures 19–21. Conversely, Figure 25 introduces the X and Y configurations.

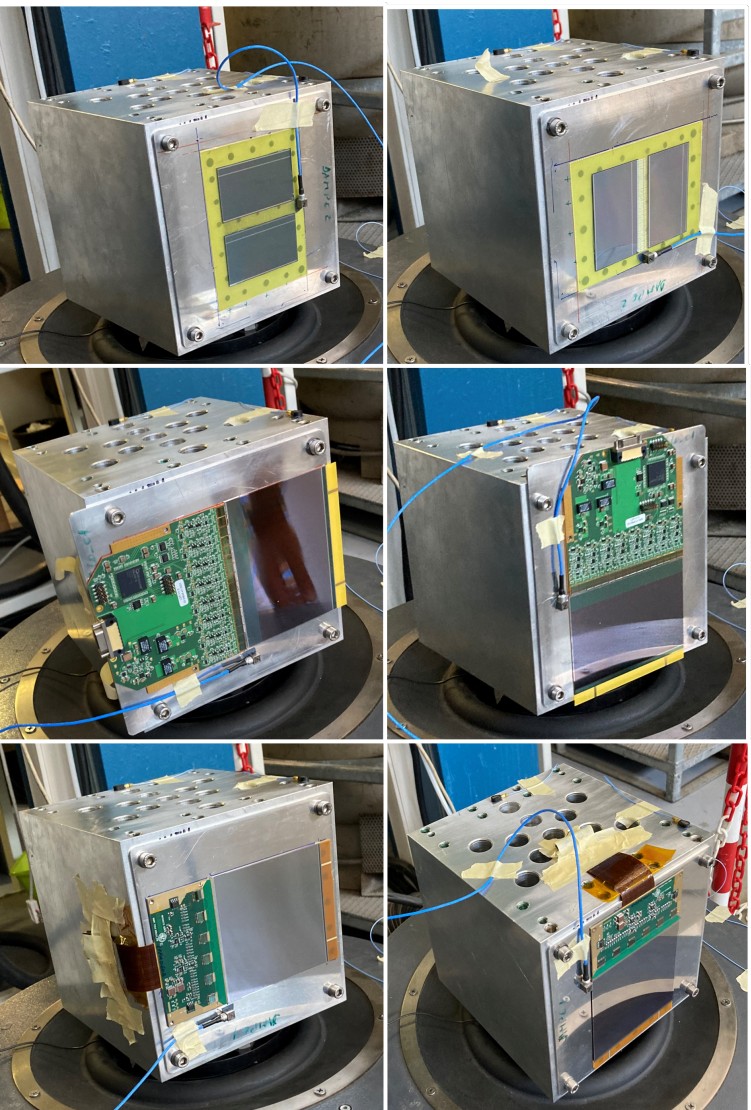

**Figure 25.** Head expander used for the X and Y test configurations.

On the topic of the specific tests, the sequence included an electrical continuity test (for samples 2 and 3) before and after each load cycle, along with a visual inspection.

Moving on to the load cycles, random and shock profiles were applied to the SUTs. The random span range was 20–2000 Hz, with a plateau between 100 Hz and 600 Hz and a ramped behavior (+3 dB, −5 dB) elsewhere. The random amplitude was dependent on the RMS. The test started from an RMS value of 14.5 g and proceeded to the maximum allowed by the shaker. Specifically, the PSD profile was not modified (same plateau). Instead,

the RMS was increased. Thus, the input was translated upwards in such a way that the profile remained the same, while the area grew (hence the RMS).

Conversely, the shock profile was based on NASA GEVs (a curve in the bi-logarithmic plane) and linearly interpolated three points: 100 Hz—81.3 g, 625 Hz—500 g, and 5000 Hz—500 g. Again, the RMS gauged the random vibration severity and ranged from 14.4 g (GEVs nominal value) to a maximum of 40 g along the Z axis and 17 g along the X and Y axes (the fixtures mass constrained the maximum acceleration). On the other hand, the shock never reached the full level provided by the GEVs, and the severity was quantified by the percentage of the full level.

Another interesting aspect to consider was the input amplification due to non-ideal mechanical connections. Here, there were two connections: one between the shaker and the metallic plate, and one between the samples' substrate and the plate. This effect was quantified using the ratio of the measured RMS and the input RMS. The amplification values (measured before the failure of sample 1) were approximately 4.4, 1.8, and 1.95 for the three test samples. The higher value for test sample 1 was explained by the accelerometer position. Indeed, for cases 2 and 3, the measurement was taken on the metallic plate, thus not taking into account the adhesive bond. Instead, in case 1, the monitoring point was on the same substrate as the detectors. Hence, it is reasonable to assume that the glue amplification effect discussed in Section 3 occurred for all the samples and that the real value on the silicon substrate was four times higher than that of the input.

Post-test functional checks and visual inspection proved successful for samples 2 and 3, while for test sample 1 the visual inspection showed a loss of adhesive integrity in correspondence with the accelerometer position (check Figure 26).

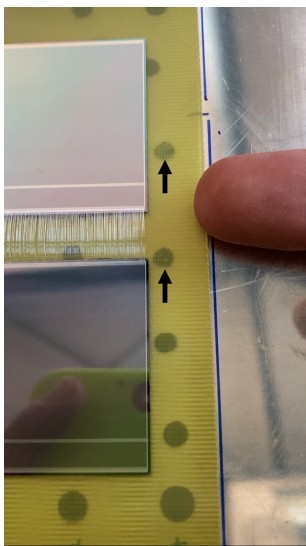

**Figure 26.** Long bond glue detachment points (marked with black arrows).

The latter event is reported here for the sake of comprehensiveness, but it was not concerning, since both the bonds and the silicon remained intact. Indeed, given the contained mass of the SUT, the accelerometer presence was not negligible and led to overtesting. Moreover, the fact that the assembly did not lose integrity when tested with an additional mass and to a very high level (RMS was more than double wrt to NASA requirements) proved the space-suitability of this technology. Additionally, the anomaly occurred after the Z random vibration. Therefore, the damaged component successfully underwent random X, Y, and shock Z.

Let us conclude this section with Table 6, summarizing the levels experienced by each test sample. Again the random intensity is quantified by the RMS (square of the integral of the PSD), while the shape of the PSD profile was not changed. Instead, the shocks were

provided as percentages of the target value (500 g), which could not be attained in the present facility.

**Table 6.** Unpotted wirebond vibration and shock test summary.

| Test Sample | Max. RMS X (g) | Max. RMS Y (g) | Max. RMS Z (g) | Max. Shock (% of NASA Reference) |
|:---:|:---:|:---:|:---:|:---:|
| 1 | 17 | 17 | 30 | 90%—450 g |
| 2 | 17 | 17 | 40 | 70%—350 g |
| 3 | 17 | 17 | 40 | 90%—450 g |

## 5. Result Summary and Conclusions

This work has covered various aspects related to spaceborne silicon detector mechanics. Namely, the mechanical characterization (extraction of elastic properties), the depiction of the gluing criticality, and wirebond strength (both in terms of wire strength and resistance to vibrations).

For the mechanical characterization, diverse AMS-02 spare detectors were tested on a three-point bending machine. Through these tests, it was possible to determine the elastic modulus, the maximum allowed stress, and the maximum allowed strain. Concurrently, another batch of AMS-02 spare detectors were weighed on a scale and measured with a caliper, leading to density estimation. A summary of the data is provided in Table 4. The characterization campaigns produced the inputs needed for the mechanical analysis of silicon detectors. Leveraging the presented data, a designer could set up a model (either static or dynamic) including silicon detectors as participating objects (not only as inert masses). This is permitted by the knowledge of elastic properties and density. Finally, failure data permit the estimation of safety factors and structural verification. As a whole, the provided data should help mechanical designers working with these detectors, by providing elastic, density, and failure data. As future developments, it would be interesting to consider the crystal orientation for a more precise assessment of mechanical properties. Although the present study provides good design inputs (in terms of mechanical properties), it could be interesting to extend the study to blank silicon wafers to give more general properties. Additionally, the three-point bending tests could be performed on operative detectors, to measure how bending affects the detector's properties. The latter investigation was not included in the present study given its focus on space applications, where the detector undergoes bending during the flight and not during its operative life. In any case, future studies could evaluate if the vibration induces a change in the silicon properties, changing its electrical properties. Section 3 details a critical mechanical aspect of silicon detector assembly: the gluing issue. Specifically, how the employment of different types of glue affects the load transfer between the substrate and the sensor. The lesson learned from a previous experiment had already resulted in changing the structural glue (from an epoxy base to a silicon base). In this document, the issue was described and further analyzed. In detail, the damping (used as a control parameter to provide qualitative information) of a nominal gluing assembly (conductive + structural glue) was compared to a non-nominal one (employing only structural glue). The negative impact of conductive glue (unavoidable and epoxy-based) was high. This study led to two conclusions: particular care should be taken when designing a sensors' gluing paths, and the conductive glue should be kept to a minimum required to ensure electrical functionality. Although each situation should be studied independently, the information contained here provides ideas and directions for preliminary mechanical analyses. The effect of different glues of the same kind is another prospect. The present research closed with a study on wirebonds (Section 4). The discussion was split in two: an initial part collecting data from a wire pull test campaign, and a vibration campaign. The first proved wirebond manufacturing quality and provided experimental data for the scientific community. The second (employing the bond manufacturing techniques validated before) proved the initial assumptions of

high-vibration tolerance of unencapsulated bonds. The successful high-level vibration tests proved that the general wirebond tolerance to space vibrations is high. The relevant result was not the proven capabilities to withstand space vibration but the successful withstanding of vibrational levels far above the standard. Indeed, the space suitability of wirebonds was proven by on-orbit operation detectors (which successfully underwent a space qualification campaign and a real space flight). As a whole, the present study shows that, not only can unencapsulated bonds withstand space vibration, but the acceleration required to break them is at least one order of magnitude (the profile was increased to the maximum of the facility and the wirebonds did not break) above the mandated standard. The present study does not disprove the use of encapsulated bonds but states that it is not necessary in space applications. Different applications may consider using this solution based on other constraints, a higher handling damage risk related to a larger volume of components for example. The present study shows that wirebonds are very resistant to vibrations. Hence, there is no need to employ performance-increase solutions such as encapsulation. On the other hand, changing the manufacturing technology negatively affects project reliably (critical in space applications). Moreover, encapsulant addition negatively impacts the mass budget (especially given the length of the ladders) and introduces another manufacturing phase, affecting the time budget. As a whole, the advantages of encapsulation do not justify the replacing of a flight-proven approach (no encapsulation) with a possibly beneficial approach with no flight heritage (to our knowledge) and with reported [34] thermo-mechanical issues.

In conclusion, this research succeeded in procuring inputs and requirements for a more accurate design of spaceborne physics experiments involving silicon detectors. The collection of information and discussion provisioned by this paper should improve the accuracy of mechanical models and pave the way for new and fascinating solutions.

**Author Contributions:** On the single authors' contributions: Conceptualization E.M., Methodology E.M., L.M., G.M. and M.P.; validation E.M., M.I. and M.P.; investigation E.M., M.I., G.M., L.M., M.C., G.S. (Gianluigi Silvestre), R.P., A.S., L.F. and G.S. (Gianluca Scolieri); software G.S. (Gianluca Scolieri); writing E.M., M.P. and A.S.; review and editing E.M.; supervision G.A., F.C. (Filippo Cianetti), C.B. and L.T.; data curation E.M.; resources F.C. (Franck Cadoux), L.M., L.T. and G.A. All authors have read and agreed to the published version of the manuscript.

**Funding:** We would like to acknowledge the European Union—NextGenerationEU under the Italian Ministry of University and Research (MUR) National Innovation Ecosystem grant ECS00000041-VITALITY for funding one of the authors.

**Conflicts of Interest:** The authors declare no conflict of interest.

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
