# Peer review of "Collection of Silicon Detectors Mechanical Properties from Static and Dynamic Characterization Test Campaigns"

_instruments, doi:10.3390/instruments7040046_

Round 1
Reviewer 1 Report
Comments and Suggestions for Authors
Dear authors, great paper and very, very interesting to all of us interested in building Si detectors.
There are a few comments that came to my mind whilst reading that I feel would be of help to the community and make this even stronger.
i) Short discussion (perhaps supported by data) of why you didn't add data on "blank" (partially processed) Si wafers for the tests. If one could establish equivalence (or not) between mechanicals for comparison in Table 1. that would be useful (time and costwise) for others performing equivalent tests.
ii) There are a large range of epoxies and Si based glues. A discussion and list of specific products tried in the research leading to this paper (and references) would be very helpful. This would, I realise, almost be a review of reviews. Issues like applications methods, thickness ranges tried (thinner layers of stiff v thicker layers, similarly for Si glues). Temperature/vacuum cycles etc. all of interest. Perhaps even patterns of application. For example are dots of stiff glue useful? All this would be great reading, but please provide more info. I suspect the details are critical.
iii) For the bond pull (pad) testing. With the level of detail you have it would be great to mention the pull tester (you do have that detail for the shake tester) you used. Also a link to the note on the details of bonding to the bond pads. would be useful. Presumably those bond tested here can be crossed referenced to some QC of larger samples. You give a number in terms of grams for average fails: it would be helpful to categorize (e.g. heel fails etc).
iv) In the discussion on space based wire bonds I was curious why there was no discussion on encapsulation of the bonds (Si glue or Epoxy). Many experiments have done this. It would be interesting to present a discussion/data on why or why not for space. It could also change the discussion on the effect of micro-wire bonds.
Comments on the Quality of English LanguageOverall, very good. There a few places where the language would do with tightening up
e.g. in Section 3.2 line 202 (detective), l 203 (Concurrently), l 207 (comprehends)
You might also look at the tenses (is/was) and make sure these are consistent and the consider the use of active/passive voices.
Reviewer 2 Report
Comments and Suggestions for Authors
The English is mostly well understandable, but with a few non-English
idiosyncrasies and some unusual or wrong use of terms.
Progressing through the report I felt more compelled to offer alternative
phrasings of the text, where the language would be used more concisely.
This mixes with my comments on reordering statements and where to add content
or remove redundant phrases - the details are given in my comments to the
authors.
Reviewer 3 Report
Comments and Suggestions for Authors
The mechanical property of silicon detctors is important. However, the paper does not show the impact of stress on the electrical property of detetors. The paper use the three-point bending test method. How the leakage current change with the stress or strain induced by the bending? Readers will expect to see the experimental results.
Round 2
Reviewer 1 Report
Comments and Suggestions for Authors
Dear author, thank you for your revision, I found the improvements really improved the content.
Comments on the Quality of English LanguageI do, however, recommend that you pass this for reading by a native reader to remove small typographical problem and phrases that tend to distract the reader. Grammatical "change" software would also help with 90% of these. So really not more than an afternoons work if you don' have anyone to help.
Reviewer 3 Report
Comments and Suggestions for Authors
I agree with the author’s response.